# The Prevalence of the Risk of Sexual Dysfunction in the First and Third Trimesters of Pregnancy in a Sample of Spanish Women

**DOI:** 10.3390/ijerph20053955

**Published:** 2023-02-23

**Authors:** Francisco Javier del Río Olvera, Yolanda Sánchez-Sandoval, Antonio Daniel García-Rojas, Susana Rodríguez-Vargas, Javier Ruiz-Ruiz

**Affiliations:** 1Institute of Biomedical Research and Innovation of Cádiz (INIBICA), University of Cadiz, 11519 Cadiz, Spain; 2Department of Pedagogy, Faculty of Education, Psychology and Sports Sciences, University of Huelva, 21002 Huelva, Spain; 3Psicología Integral SusanaRodVAR, 41018 Seville, Spain; 4Centro de Salud San Benito, 11407 Jerez de la Frontera, Spain

**Keywords:** sexual dysfunction, pregnancy, depression, female sexuality, FSFI

## Abstract

Sexuality is a central aspect for all human beings. Research into the prevalence of sexual dysfunction in pregnant Spanish women is scarce. The aim of this work is to examine the prevalence of risk of sexual dysfunctions in pregnant Spanish women and determine in which trimester the greatest difficulties in sexual response occur. The sample consisted of 180 pregnant Spanish women, with an average age of 32.03 years (SD = 4.93). The participants completed a questionnaire for socio-demographic data, as well as the female sexual function index, the state/trait depression inventory, and the dyadic adjustment scale. The results indicate that the percentage of women with a risk of sexual dysfunction was 65% in the first trimester and 81.11% in the third trimester. Likewise, the highest score on the depression questionnaire was in the third trimester, and the couple’s relationship also improved in the third trimester. To improve women’s sex lives during pregnancy, it is recommended to increase sexual education and information for both pregnant women and their partners.

## 1. Introduction

Sexuality is a core aspect for all humans [1,2]. The Pan American Health Organization and the World Health Organization define sexual health as “the experience of the ongoing process of achieving physical, psychological, and sociocultural well-being related to sexuality. Sexual health is observed in the free and responsible expressions of sexual capacities that lead to harmonious personal and social well-being, thus enriching individual and social life” [3] (p. 6). Sexual health reflects the physical, mental, and emotional health of an individual [4]. Sexual inactivity has been found, for example, to be associated with health problems, including hypertension, diabetes, and cardiovascular problems [5,6]. Increased sexual activity has also been associated with positive indicators of physical, social, and emotional health [7].

During pregnancy, a woman undergoes a series of gradual physiological changes in her body at all levels as her body adapts to the gestation process. Changes occur at the cardiovascular, digestive, pulmonary, hematological, and endocrine levels [8]. More specifically, it should be noted that, in physical terms, during the first trimester, vomiting and nausea may occur, as well as breast tenderness, among other symptoms [9]. Psychologically, women experience changes in their self-image, especially from the second and third trimesters onwards [9]. In the case of sexuality during pregnancy, several authors point out the benefits of sexual activity, such as facilitating childbirth, improving fetal well-being, and promoting harmony in the couple [10,11,12].

Hatzimouratidis and Hatzichristou [13] defined sexual dysfunction as diminished feelings of sexual interest, the absence of desire, sexual thoughts, or fantasies, and a lack of responsive desire. Sexual dysfunction affects both men and women, although men have been studied more in the literature [14]. This dysfunction may have a medical or psychological cause or be due to both causes at the same time. The different psychological causes may include relationship difficulties, anxiety, fear, depression, and cultural conditioning factors, among others [15,16]. Regarding sexual response during pregnancy, previous studies indicate that it is influenced by biological, psychological, sociocultural, and religious factors [17,18,19,20]. Factors that have been reported to favor the avoidance of sexual encounters include fears and myths about sexuality in pregnancy, such as harming the fetus, causing a miscarriage, bleeding, infections, or causing premature birth [20,21,22,23,24,25]. Likewise, some studies state that couples go through a period of adjustment that can cause changes in intimacy and the frequency of sexual activity, with these changes even affecting the postpartum period and breastfeeding [26,27,28].

Data on the prevalence of sexual dysfunction during pregnancy are varied, ranging from studies reporting 31.2% [29] to those reporting 87.4% [30]. Leite et al. [29] carried out a study on the prevalence of sexual dysfunction in pregnant women in each trimester of pregnancy, both in adolescent and adult women. The results of these authors indicate that adolescent women had a percentage of dysfunctional scores in the first trimester of 40.8%, a percentage in the second trimester of 31.2%, and a percentage in the third trimester of 63.2%. In turn, adult women presented with scores of 46.6% dysfunction in the first trimester, 34.2% in the second trimester, and 73.3% in the third trimester. As can be seen, the periods with the highest dysfunctionality percentages are the first and third trimesters, regardless of the age of the pregnant woman. Likewise, Ninivaggio et al. [31] also found higher percentages of dysfunctionality in the third trimester (57%) than in the other two, although no significant differences were found between the second and first trimesters (36.8% and 36.3%, respectively). On the other hand, it is necessary to point out data on the prevalence of sexual dysfunction from studies that do not distinguish between the different trimesters of pregnancy. Naldoni et al. [32] reported a 61% prevalence in a Brazilian sample. Daud et al. [33] in a study on the prevalence of sexual dysfunction during pregnancy, reported 81%. Olivares-Noguera et al. [34], in another prevalence study, in this case carried out in Colombia, found a prevalence of 37.7%. Finally, Karakas et al. [30], in a study conducted during the COVID-19 pandemic, reported a prevalence in pregnant women of 87.4%. Although these works show very different information in some cases, the main advantage of these studies is that the same measurement instrument, the female sexual function index [35], was used, which allows comparisons among the data.

According to Leite et al. [29], despite the growing body of research on the prevalence of female sexual dysfunction, there is insufficient data on the prevalence in women during pregnancy. This is particularly noteworthy in the Spanish setting, as there are few studies that include the prevalence of sexual dysfunction in pregnant women in this population. This may be due to the taboo still represented by sexuality in the Spanish population in general and in the case of sexuality during pregnancy in particular. For this reason, the aim of this research is to describe the prevalence of the risk of sexual dysfunction in a group of pregnant Spanish women, while also analyzing the relationship between different aspects of sexual response and sexuality. The hypotheses are as follows: there will be more difficulties in sexual response in the third trimester than in the first, the couple’s relationship will improve as the time of delivery approaches, the trimester of pregnancy in which they score higher in depression will also have a higher percentage of difficulties in sexual response.

## 2. Materials and Methods

### 2.1. Participants

Sampling took place between December 2016 and March 2018 and 180 pregnant women participated in the study. The mean age of the women was 32.03 years, with a standard deviation of 4.93. In 56.11% of women, it was their first pregnancy, 37.22% already had one child, 6.11% had two children, and 0.56% had three. At the time of data collection in the first trimester of gestation, women were on average at 8.31 (SD = 2.07) weeks of gestation, and in the third trimester, they were at 33.29 weeks of gestation (SD = 3.06). The remaining sociodemographic information is presented in Table 1.

### 2.2. Instruments

An ad hoc instrument was used to collect the sociodemographic data of the women taking part. The questionnaire asked about age, sexual orientation, number of children, type of previous birth (if any), marital status, and whether they were working at the time of participation in the study. They were also asked for authorization to access their clinical details (blood testing).

Female Sexual Function Index (FSFI) [35]. The version adapted to Spanish was used [36]. The index consists of nineteen questions that are grouped into six domains: desire, arousal, lubrication, orgasm, satisfaction, and pain; each question has 5–6 answer alternatives, with a score ranging from 0 to 5 for each question. In order to answer the questionnaire, it is necessary to have had sexual activity in the last 4 weeks. The score for each domain is multiplied by a factor and the final result is the arithmetic sum of the domains; the higher the score, the better the perceived sexuality. The cut-off point for defining the risk of sexual dysfunction is a score of fewer than 26.5 points. In the present research, the women participants completed the questionnaire in the first and third trimesters of pregnancy, with Cronbach’s alpha values of 0.994 and 0.995, respectively.

State/Trait Depression Inventory (ST/DEP) [37]. The test aims to identify the degree of impairment (state) and frequency of occurrence (trait) of the affective component of depression. This questionnaire consists of twenty items distributed in two scales: trait and state, each with ten items. The total score for each scale is obtained by adding the scores of the items, ranging from 10 to 40. In the present investigation, the women participants completed the questionnaire in the first and third trimesters of pregnancy, with Cronbach’s alpha values of 0.748 and 0.775, respectively.

Dyadic Adjustment Scale (DAS) [38]. The reduced version of thirteen items was used, adapted, and validated by Santos-Inglesias et al. [39], to be answered on a Likert-type scale, with 5–6 response options. In addition to a global scale, it assesses three factors: consensus, satisfaction, and cohesion. A higher score indicates a better fit in the couple. In the present research, the women participants completed the questionnaire in the first and third trimesters of pregnancy, with Cronbach’s alpha values of 0.768 and 0.806, respectively.

### 2.3. Procedure

The sample was collected by non-probability incidental sampling. This can be considered a time series study [40]. Three public health centers from an urban area of the province of Cádiz took part. The participating women were contacted at the reference health center when they attended for pregnancy control. When they went for a pregnancy check, they were given information about the research and asked to volunteer. They were asked to fill in the research questionnaires during the first and third trimesters of pregnancy. The inclusion criteria consisted of attending pregnancy control at the health centers participating in the research, providing samples in at least the first and third trimesters, and giving consent to access the data from the analyses. Confidential patient data were processed only by the principal researcher, and each participant was coded to ensure privacy of the information. This research was approved by the Research Ethics Committee of the province of Cadiz.

## 3. Results

First, descriptive data were calculated for each of the questionnaires in both the first and third quarters (see Table 2). It can be observed that scores on the female sexual function questionnaire are lower in the third trimester than in the first trimester, which would be indicative of a higher frequency of the risk of sexual dysfunction in the third trimester. The DAS score indicates better adjustment in the third trimester than in the first trimester. Scores on the depression questionnaire are higher in the third trimester than in the first trimester, indicating a greater likelihood of depressive symptomatology in the third trimester.

To check for significant differences between the scores in both trimesters, and to help decide whether to use parametric or non-parametric tests, the Kolmogorov–Smirnov normality test was performed. The result of the normality test recommended the use of non-parametric tests (*p* < 0.001). Thus, the decision was taken to use the Wilcoxon test for paired samples to test for differences between the scores for the two trimesters.

The data taken from the FSFI were transformed according to their risk of developing sexual dysfunction in women, taking into account that, in the general scale, the cut-off point is 26.5, and in the other scales, it is 3.6 [35,41]. The data are presented in Table 3 (see Table 3).

Finally, the relationship between the different domains of the FSFI was analyzed using the HJ-Biplot [42]. Through this analysis, it is possible to observe the high relationship between pain and lubrication, as their vectors appear almost united. Similarly, arousal and orgasm are also closely related to pain and lubrication. On the other hand, desire and satisfaction are more closely related to each other than to the other FSFI domains (Figure 1).

## 4. Discussion

The study found a first-trimester prevalence of 65% in the risk of sexual dysfunction (ranging across FSFI domains from 63.34% to 66.11%) and a third-trimester prevalence of 81.11% (ranging across FSFI domains from 78.89% to 88.22%). The prevalence was highest in the third trimester, with orgasm being the domain most affected.

The prevalence data are similar to those reported by Daud et al. [33], who found an 81% prevalence of sexual dysfunction in 100 pregnant women in Malaysia. An essential aspect of the prevalence of sexual dysfunction is the trimester in which the sexual response is analyzed, as a large part of the oscillations in the percentages found in studies on sexual function in pregnant women are related to this variable. The research carried out by Daud et al. [33] took place in the third trimester, which explains the high prevalence found. Comparing their data with those of the present study, it is noted that they are similar to those obtained in the third trimester (81.11%) but are relatively high when compared to the data for the first trimester (65%). A contrast was found in the study by Olivares-Noguera et al. [34], carried out in Colombia in 98 pregnant women. The authors found a sexual dysfunction prevalence of 37.7%, higher than that observed in the present study. Here, the main difference that may justify contrasting the data is the number of previous children. Several studies point to the importance of taboos and lack of information on sexuality during pregnancy, which often lead to fear of harming the baby and of premature birth [43,44,45]. In the work by Olivares-Noguera et al. [34], the percentage of women who had had a previous pregnancy amounted to 70.41%, so these were women who had previous experience and could have verified the untruthfulness of myths about sexuality and pregnancy. In the present study, the number of women who had not had a previous pregnancy was 60%. As this is their first pregnancy, these women may have a lack of information and previous experience and may behave according to myths about sexuality and pregnancy.

Regarding the relationship between the different FSFI domains, it is worth noting the high relationship between pain and lubrication and the relationship, to a lesser extent, with arousal and orgasm. Finally, desire and satisfaction seem to form a separate group. Discovering the relationship between the domains of the FSFI will enable us to infer the need to assess a given domain, even if no distress is reported, simply because distress is indicated with the related domain.

Regarding the first hypothesis of the study, which indicated that there will be more difficulties in sexual response as pregnancy progresses, the data confirm this hypothesis. A higher prevalence of the risk of sexual dysfunction is observed in both individual and total FSFI domains in the third trimester compared to the first trimester. This conclusion is similar to that found in previous studies [2,29,31,41]. According to various authors, the reasons for this worsening of sexual response may include harming the fetus and its development, lack of information, feeling less attractive, and physical fatigue [41,43,46]. It is remarkable that difficulties in sexual response do not stem from physical difficulties, but that in the age of information and communication, it is the lack of information and the belief in certain taboos that continue to negatively influence sexual response. This should be an indicator for improving sexual information for pregnant women, from the point of care in health centers and hospitals.

The second hypothesis states that the couple’s relationship will improve as the time of delivery approaches, and the data confirm the hypothesis. The benefits of accompanying a partner during pregnancy and childbirth are well known [47], which encourages the rapprochement of the couple throughout these stages. There is now a positive perception of partner involvement throughout the course of pregnancy and especially during childbirth as emotional support for the mother [48]. Construction of the paternal bond with the child is described as a process that begins at gestation [49], which facilitates and promotes dyadic adjustment in the couple, as observed in the outcomes of this research.

The third hypothesis states that the trimester of pregnancy in which a higher depression questionnaire score is reached will also result in a higher percentage of difficulties in sexual response. The trimester in which the highest mean depression questionnaire scores are found is the third trimester, as well as when the highest percentages of difficulties in sexual response are reported, so the data confirm the hypothesis. Several studies have found an increase in depressive symptomatology during pregnancy [50,51,52]. In turn, Bogren [53] noted that sexual desire and satisfaction were affected by depressive moods. Finally, both Chang et al. [54], and DeJudicibus and McCabe [19] reported that depressive symptomatology was an important predictor of reduced sexual desire and sexual satisfaction.

This work has implications in the clinical field that need to be highlighted. The results indicate that those women who were more at risk of sexual dysfunction during pregnancy also showed a greater presence of depressive symptoms. Improving sexual response during pregnancy will have benefits both for the woman (strengthening the pelvic floor, improving self-esteem, greater personal satisfaction, and fewer depressive symptoms, among others) and for the couple’s relationship (greater intimacy and mutual satisfaction).

This study is not without limitations. Firstly, non-probability incidental sampling was used, which may limit the generalisability of the results. Another possible limitation is the failure to assess possible sexual dysfunction or attitudes towards sexuality in couples, as this may contribute to the presence of sexual dysfunction in women. Finally, it should be noted that women’s sexual function was also not assessed prior to pregnancy, as some women may have had sexual dysfunction beforehand. The main strength of the study is the use of the FSFI questionnaire, as it is a survey widely used in research and allows comparison of data with other research. For future research, the variables to be studied should be expanded considerably in order to better analyze the factors contributing to the presence of sexual dysfunction in pregnant women and to determine, for example, whether they are biological or social in nature, and to what extent they may affect relationships, including assessing how the use of addictive substances by the partner may affect the pregnant woman. Similarly, it is recommended that a larger study be conducted to allow greater generalization of the results, investigating the relationship that may exist between sexual dysfunction and experience of previous pregnancies, examining the sexual response prior to pregnancy, and analyzing the sexual response and attitudes of couples.

## 5. Conclusions

This research shows the prevalence of the risk of sexual dysfunction in a sample of Spanish women in the first and third trimesters of pregnancy, as well as the link with depression and their relationship as a couple. In general terms, the prevalence of risk of sexual dysfunction was found to be higher in the third trimester than in the first trimester. The improvement in the couple’s relationship in the third trimester compared to the first was also highlighted. Finally, it was also observed that the depression questionnaire score is higher in the trimester in which there are greater difficulties in sexual response. To improve women’s sex lives during pregnancy, it is recommended to increase the sexual education and information available for both pregnant women and their partners, with a view to debunking possible myths and/or false beliefs. Likewise, it would also be advisable to improve the social conditions of pregnant women, for example by reducing their working hours during the third trimester.

## Figures and Tables

**Figure 1 ijerph-20-03955-f001:**
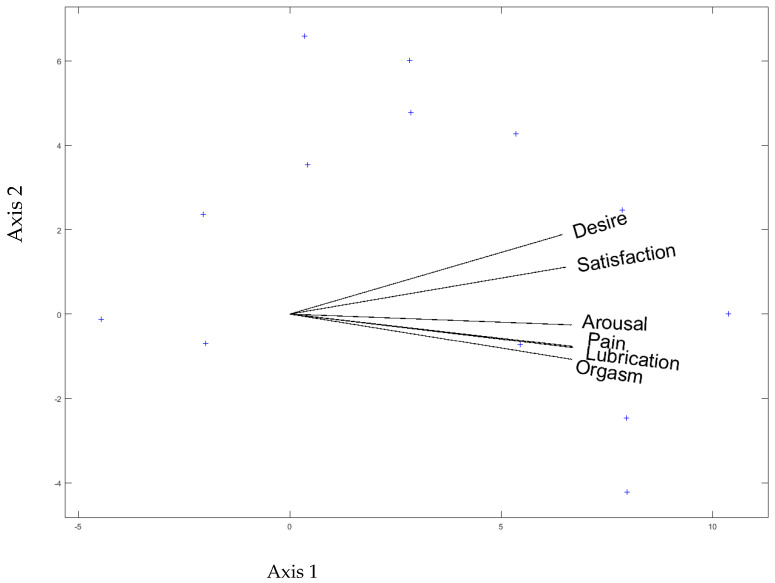
HJ-Biplot representation.

**Table 1 ijerph-20-03955-t001:** Socio-demographic data.

Variables	*n*	1st Trimester	3rd Trimester
Marital Status			
Single	67	37.2%	37.2%
Married	105	58.3%	58.3%
Widow	1	0.6%	0.6%
Domestic partnership	7	3.9%	3.9%
Currently working			
No	74	41.1%	41.1%
Yes	106	58.9%	58.9%
Previous birth			
No previous birth	108	60.0%	60.0%
Natural birth	54	30.0%	30.0%
Cesarean	18	10.0%	10.0%
Sexual orientation			
Heterosexual	179	99.4%	99.4%
Homosexual	1	0.6%	0.6%

**Table 2 ijerph-20-03955-t002:** Mean scores per questionnaire and comparison of differences (*n* = 180).

	First Trimester			Third Trimester					
Variables	Mean	SD	Min	Max	Mean	SD	Min	Max	*p*
Desire	2.46	3.21	1.2	5.4	1.21	2.29	1.2	6	**
Arousal	5.63	7.52	1.2	6	3.13	6.21	0.9	6	**
Lubrication	5.24	8.38	1.2	6	3.56	7.10	0.9	6	**
Orgasm	4.03	5.68	1.2	6	2.56	5.23	1.2	6	**
Satisfaction	4.42	5.94	1.2	6	2.88	5.43	1.2	6	**
Pain	4.03	5.68	1.2	6	2.68	5.28	0.8	6	**
FSFI total	27.07	23.62	19	36	16.01	18.32	10	36	**
DAS	12.21	22.42	6	52	21.62	25.77	8	55	**
ST/DEP trait	5.44	9.40	0	25	8.86	10.75	0	30	**
ST/DEP state	5.28	9.14	0	25	8.77	10.68	0	29	0.092

Note: ** *p* < 0.01; DAS, dyadic adjustment scale; ST/DEP, state–trait depression inventory; FSFI, female sexual function index.

**Table 3 ijerph-20-03955-t003:** Frequency and percentage of dysfunctionality risk by FSFI scales.

	First Trimester		Third Trimester	
Variables	*n*	%	*n*	%
Desire	115	63.89	146	81.11
Arousal	114	63.34	145	80.56
Lubrication	116	64.45	147	81.67
Orgasm	117	65.00	148	88.22
Satisfaction	115	63.89	142	78.89
Pain	119	66.11	146	81.11
FSFI total	117	65.00	146	81.11

Note: *n* = frequency of the risk of sexual dysfunctionality; % = percentage of the risk of sexual dysfunctionality; FSFI = female sexual function index.

## Data Availability

The data are not publicly available due to privacy.

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
