# Peer review of "The Prevalence of the Risk of Sexual Dysfunction in the First and Third Trimesters of Pregnancy in a Sample of Spanish Women"

_ijerph, 2023, doi:10.3390/ijerph20053955_

Round 1

Reviewer 1 Report

In materials and methods - Participants: you write that 180 pregnant women were examined, there is no information on how many of them were excluded from the study based on the inclusion and exclusion criteria.

There is no information on how many women did not complete FSFI due to lack of sexual activity in the last 4 weeks - as assumed by FSFI, additionally this information is missing in the description of the tool.

The values of the standard deviation (SD) are very high, as evidenced by the large range of results obtained between the surveyed women.

in table 3, the N value is given, which does not match in terms of filling out the forms, on what basis it was determined - by default, if I have a FSFI total = 117 N, then in each domain there should also be N = 117. Similarly in the 3rd trimester of pregnancy. A satistic analysis to be re-analyzed by the number of participants.

Figure 1.1 - it is illegible to be corrected, a different graphical version of presenting these results can be used.

Results:Table 2: the value of N and the minimum and maximum values obtained by the respondents are not given, which is important for the interpretation of the results and their correct presentation.

Author Response

Thank you for your comments.

In materials and methods - Participants: you write that 180 pregnant women were examined, there is no information on how many of them were excluded from the study based on the inclusion and exclusion criteria.

We are grateful for the comment. However, no record was kept of the women who did not wish to participate in the study, or who did not attend pregnancy monitoring at the relevant time, so this information was not included.

There is no information on how many women did not complete FSFI due to lack of sexual activity in the last 4 weeks - as assumed by FSFI, additionally this information is missing in the description of the tool.

Thank you for the comment. However, all women who voluntarily participated in the study reported having had sexual intercourse within the previous 4 weeks. Additional information has been included in the description of the instrument. 

The values of the standard deviation (SD) are very high, as evidenced by the large range of results obtained between the surveyed women.

Thank you very much for the comment. Your observation will be taken into account

 In table 3, the N value is given, which does not match in terms of filling out the forms, on what basis it was determined - by default, if I have a FSFI total = 117 N, then in each domain there should also be N = 117. Similarly, in the 3rd trimester of pregnancy. A statistical analysis to be re-analysed by the number of participants.

Thank you for your comment. In Table 3, the “n” indicated does not refer to the total number of participants, but only to the percentage of women at risk of dysfunction, according to the scales, and those at risk in the whole questionnaire.

Figure 1.1 - It is illegible to be corrected, a different graphical version of presenting these results can be used.

Thank you for your comment. The program used to create the graph does not allow its modification. Nevertheless, we have tried to improve it by adding the text for the axes and increasing the image size.

Results: Table 2: the value of N and the minimum and maximum values obtained by the respondents are not given, which is important for the interpretation of the results and their correct presentation.

Many thanks for the comment. The table has been revised and the data included, the errors have been corrected.

Author Response

Thank you for your comments.

The topic taken up by the Authors is significant in terms of women's sexual health, as sexual dysfunctions occurring among them can have a destructive impact on various spheres of their functioning, such as their relationship with their life partner. In fact, a woman's sexual dysfunction can affect the level of support she receives from her partner both during pregnancy and after the child is born. Another important reason is that many cultures still maintain social stereotypes that confine the pregnant woman to a maternal role, thus devaluing her sexual needs.

Introduction: contains a coherent argument; it would be useful to discuss more extensively the benefits and risks of sexual intercourse for the pregnant woman and for the sustainability of her marriage or partnership (regarding physical and mental as well as emotional health). A further elaboration on the determinants of sexual dysfunctions occurring during pregnancy would be relevant. At this point, it would also be worthwhile to highlight the contribution of social stereotypes that narrow a pregnant woman down to the role of mother, and devalue her sexual needs.

Many thanks for the suggestion. We have modified the Introduction.

Materials and Methods: proper research design, appropriate hypotheses; please clarify the description of the study area: describe in how many health centres and in which locations the study was conducted. Whether they were urban or rural clinics, whether free or paid (e.g. private clinics for the elite). Note that this is relevant due to the fact that women's social status may be an important intermediary variable associated with their sexual satisfaction levels.

Thank you very much for the suggestion. The information has been included.

In the present study with the Female Sexual Function Index (FSFI) tool, Cronbach's alpha values are significantly higher than in previous cited studies (Rosen RC, Brown J, Heiman S, Leiblum SR, Meston CM, Shabsihg R, et al. The Female Sexual Function Index (FSFI): A Multi-297 dimensional Self-Report Instrument for the Assessment of Female Sexual Function. J Sex Marital Ther. 2000 Apr;26(2):191-208.). Please explain if the current research involved the Author's adaptation of the tool, or if an existing Spanish-language version was used (please provide the source).

Thanks for the suggestion. The reference to the Spanish adaptation is now included.

There is no need to present percentages to the hundredths in Table 1 for a sample of 180 people, decimal parts are sufficient (e.g. 5.8%).

Thank you for your comment. This has been modified.

Please provide the number of the Bioethics Committee decision.

The research ethics committee in Cadiz does not provide a number when approving the study. The (unnumbered) approval certificate from the committee has been forwarded to the journal editor. Should the reviewer wish, we can also provide them with a copy.

Discussion: Very clear and cogent discussion. Findings of own research precisely juxtaposed with other authors' research. It would be worth remarking that future research should be considerably expanded to include the issue of the determinants of women's sexual dysfunctions (also attempts to determine whether they are of a biological or rather a social nature; to what extent they are related to, for instance, relations in the relationship, the partner's sexual dysfunctions, the partner's abuse of psychoactive substances, the partner's attitude towards pregnancy and his stereotypical beliefs about the pregnant woman); another key question is to assess to what extent these dysfunctions affect the quality of intimate relationships (marriages and partnerships of women).

We are very grateful for the suggestions, which are now included in the article.

Conclusions: The Authors recommend that the quality of women's sexual lives should be improved by the intensification of education of women and their partners. However, I would also suggest making further suggestions for systemic actions [e.g. towards the improvement of women's social conditions, which may indirectly decrease the magnitude of sexual dysfunctions in women (e.g. reducing women's working hours during the 3rd trimester of pregnancy)].

We are very grateful for the suggestions, which are now included in the article.

Reviewer 3 Report

The aim of the present manuscript, as stated, was to examine the prevalence of sexual dysfunctions in Spanish pregnant women and determine in which trimester the greatest difficulties in sexual response occur. The authors did so with scientifically sound methodology and appropriate psychometric tests. However, at the current stage, the strength of the present manuscript should be improved.

The percentage of single women in each trimester should be stated, as it could potentially affect all the scales, particularly the DAS. Alike, the percentages of primiparous and multiparous women in each trimester too. Indeed, it is likely that differences exist in terms of sexual dysfunctions, depression and adjustment in primiparous women compared to multiparous women, and they should be eventually addressed. If not possible, this should be listed in the limitations section.

The current research focus on the first trimester and third trimesters only. Because the second trimester is not included, calling it “a progression” (line 77) is not appropriate, also considering the fact that other literature that discovered that there is not such a linear progression of sexual dysfunctions has been mentioned by the authors (lines 51-53). 

The clinical implication of the present manuscript should be better discussed. Why the authors consider their research to be important? Perhaps expanding the discussion of the relationship between the different psychometric tools would be valuable, in this sense.

Finally, some minor editing issues:

- line 155 activation (you mean arousal?)
- 209 reapproAchment
- some sentences are too long (lines 120-123) or contain repetitions (lines 197-199, perhaps “harming the foetus and its development” may sound cleaner?), hence I suggest rephrasing.

Author Response

Thank you for your comments.

The aim of the present manuscript, as stated, was to examine the prevalence of sexual dysfunctions in Spanish pregnant women and determine in which trimester the greatest difficulties in sexual response occur. The authors did so with scientifically sound methodology and appropriate psychometric tests. However, at the current stage, the strength of the present manuscript should be improved.

The percentage of single women in each trimester should be stated, as it could potentially affect all the scales, particularly the DAS. Alike, the percentages of primiparous and multiparous women in each trimester too. Indeed, it is likely that differences exist in terms of sexual dysfunctions, depression and adjustment in primiparous women compared to multiparous women, and they should be eventually addressed. If not possible, this should be listed in the limitations section.

Thank you for your comments. The percentage data requested by the reviewer are in Table 1. They are totals and do not vary between trimesters of pregnancy.

The current research focus on the first trimester and third trimesters only. Because the second trimester is not included, calling it “a progression” (line 77) is not appropriate, also considering the fact that other literature that discovered that there is not such a linear progression of sexual dysfunctions has been mentioned by the authors (lines 51-53). 

Many thanks for the suggestion. The text has been modified.

The clinical implication of the present manuscript should be better discussed. Why the authors consider their research to be important? Perhaps expanding the discussion of the relationship between the different psychometric tools would be valuable, in this sense.

Many thanks for the suggestion. Now included in the text.

Finally, some minor editing issues:

- line 155 activation (you mean arousal?)
- 209 reapproAchment

My apologies to the reviewer, but I do not understand what they wish to point out with the above comment.

- some sentences are too long (lines 120-123) or contain repetitions (lines 197-199, perhaps “harming the foetus and its development” may sound cleaner?), hence I suggest rephrasing.

We are very grateful for the suggestions, which have been included in the article.

Reviewer 4 Report

This is an interesting and well-written paper about a relevant topic and that can help to spark debate and further discussion within the academic community, leading to the advancement of knowledge and understanding in the field. The methodology used is correct and well-detailed, as this provides a foundation for the validity and reliability of the results.

There are, however, two elements of the paper that need improvement.

First, the paper would benefit from a more detailed theoretical framework providing a conceptual basis for the study and setting the context for the research questions and hypotheses. The same goes for the discussion and conclusion sections that need some deepening of the arguments.

Second there are some exaggerated interpretation of the results obtained by the data collection instruments namely: i) “sexual dysfunction”; ii) “depression score”. This two concepts are used abusively along the paper, and I strongly recommend its revision.

Dysfunction refers to a failure or malfunction of a part or system, resulting in a diminished or impaired function. Sexual dysfunction, in a medical or psychological context, is a condition that depends on a diagnosis.  The instrument used by the authors (Female Sexual Function Index - FSFI) doesn’t set a diagnosis; it provides, at most, a risk of sexual dysfunction. So I recommend, all along the paper (including the title), to avoid the term “sexual dysfunction”, replacing it for another (for example: “risk of sexual dysfunction”).

The same happens with the term “depression score” used along the paper. Depression is a pathological condition diagnosed by medical means. So, the State/Trait Depression Inventory (ST/DEP) measures depressive symptoms not depression.

Despite my suggestions, this is a work of excellent quality that deserves scientific dissemination

Author Response

Thank you for your comments.

This is an interesting and well-written paper about a relevant topic and that can help to spark debate and further discussion within the academic community, leading to the advancement of knowledge and understanding in the field. The methodology used is correct and well-detailed, as this provides a foundation for the validity and reliability of the results.

There are, however, two elements of the paper that need improvement.

First, the paper would benefit from a more detailed theoretical framework providing a conceptual basis for the study and setting the context for the research questions and hypotheses. The same goes for the discussion and conclusion sections that need some deepening of the arguments.

Many thanks for the suggestion. The Introduction, Discussion and Conclusion sections have been modified in consequence.

Second there are some exaggerated interpretation of the results obtained by the data collection instruments namely: i) “sexual dysfunction”; ii) “depression score”. This two concepts are used abusively along the paper, and I strongly recommend its revision.

Dysfunction refers to a failure or malfunction of a part or system, resulting in a diminished or impaired function. Sexual dysfunction, in a medical or psychological context, is a condition that depends on a diagnosis.  The instrument used by the authors (Female Sexual Function Index - FSFI) doesn’t set a diagnosis; it provides, at most, a risk of sexual dysfunction. So I recommend, all along the paper (including the title), to avoid the term “sexual dysfunction”, replacing it for another (for example: “risk of sexual dysfunction”).

The same happens with the term “depression score” used along the paper. Depression is a pathological condition diagnosed by medical means. So, the State/Trait Depression Inventory (ST/DEP) measures depressive symptoms not depression.

We are grateful for the suggestion. The term “sexual dysfunction” has been replaced by “risk of sexual dysfunction”, and the term “depression score” by “depression questionnaire score”.

Despite my suggestions, this is a work of excellent quality that deserves scientific dissemination. 

Round 2

Reviewer 1 Report

Work ready for publication. All comments from reviewers have been taken into account.

Author Response

Many thanks to the reviewer. Thank you for considering the manuscript ready for publication. 

Reviewer 3 Report

Thanks to the authors for having addressed my comments. However, the following response appears not quite clear: "the percentage data requested by the reviewer are in Table 1. They are totals and do not vary between trimesters of pregnancy"

There are 67 single women. How many of them are in the 3rd trimester? how many in the 1st? How many primiparous women are in the 3rd trimester? How many in the first?

I suggest that if a substantial percentage of single women, perhaps even primiparous, is in the third trimester, this could definetely affect the total results.

Also, how single women could fill in the DAS? Was this taken into consideration?

Thanks.

Author Response

Many thanks to the reviewer for his comments. Table 1 has been modified to include the reviewer's suggestions. 
Regarding the percentage of single primiparous women in the third trimester, it is the same as in the second trimester, so it does not affect the results. They are the same women taken in the two trimesters indicated. 
Single women were asked if they had a partner in order to participate in the study. All single female participants had a partner at the time of participation in the research.